# Turning Seashell Waste into Electrically Conductive Particles

**DOI:** 10.3390/ijms23137256

**Published:** 2022-06-29

**Authors:** Stefanie Gärtner, Angelina Graf, Carla Triunfo, Davide Laurenzi, Stefan M. Schupp, Gabriele Maoloni, Giuseppe Falini, Helmut Cölfen

**Affiliations:** 1Department of Chemistry, Physical Chemistry, University of Konstanz, Universitätsstrasse 10, Box 714, D-78457 Konstanz, Germany; stefanie.gaertner@uni-konstanz.de (S.G.); angelina.graf@uni-konstanz.de (A.G.); 2Dipartimento di Chimica “Giacomo Ciamician”, Alma Mater Studiorum Università di Bologna, via F. Selmi 2, 40126 Bologna, Italy; carla.triunfo2@unibo.it (C.T.); giuseppe.falini@unibo.it (G.F.); 3Finproject S.p.A., Plant Ascoli Piceno, Via Enrico Mattei, 1—Zona Ind.le Campolungo, 63100 Ascoli Piceno, Italy; d.laurenzi@finproject.com (D.L.); g.maoloni@finproject.com (G.M.); 4Department of Physics, University of Konstanz, Universitätsstraße 10, Box 714, D-78457 Konstanz, Germany; stefan.schupp@uni-konstanz.de

**Keywords:** seashells, waste, clam, calcium carbonate, biomaterials, conductivity, polypyrrole

## Abstract

Biomaterials such as seashells are intriguing due to their remarkable properties, including their hierarchical structure from the nanometer to the micro- or even macroscopic scale. Transferring this nanostructure to generate nanostructured polymers can improve their electrical conductivity. Here, we present the synthesis of polypyrrole using waste seashell powder as a template to prepare a polypyrrole/CaCO_3_ composite material. Various synthesis parameters were optimized to produce a composite material with an electrical conductivity of 2.1 × 10^−4^ ± 3.2 × 10^−5^ S/cm. This work presents the transformation of waste seashells into sustainable, electronically conductive materials and their application as an antistatic agent in polymers. The requirements of an antistatic material were met for a safety shoe sole.

## 1. Introduction

Seashells are composite materials based on calcium carbonate (CaCO_3_) and macromolecules. They stand out due to their fascinating properties such as hardness, complex morphologies, and optical properties. The astonishing mechanics are due to first, their hierarchical structure from the nanoscale to the macro scale, and second, their so-called “brick-and-mortar” structure, which in the case of nacre consists of alternating layers of CaCO_3_ separated by organic matrixes. In this case, CaCO_3_ is in the form of pseudo-hexagonal, polygonal, or rounded aragonite platelets, which make up ~95 wt% of the nacre. The organic matrix and water make up the remaining ~5 wt% of the material and the former is composed of polysaccharides and proteins [1,2,3,4].

Each year, seven million tons of seashells are generated as waste in the fishing industry. In developing countries, this waste is dumped into the sea, which generates severe impacts on the environment [5]. These impacts include, for instance, pollution of coastal fisheries, damage of the natural landscape and health/sanitation problems [6]. Therefore, the cement industry, for instance, uses seashells as an active aggregate substitute to reduce these problems and to take potential advantage of the mechanical properties, including workability, compressive strength, and tensile strength of seashells [7]. There are several other applications for waste seashells, including water absorption and use as fillers in polymer composites [8,9,10,11]. In this paper, we present the conversion of waste seashells into sustainable, electronically conductive materials. One way to achieve this goal is to synthesize electrically conductive polymers using seashells ground into particles as a template. Electrically conductive polymers are conjugated polymers due to their conjugated double bonds. This leads to a π-electron system. These polymers become electrically conductive either by removing electrons (oxidation) or inserting them into the polymer matrix. This process is called doping and can be achieved, for instance, by exposing the polymer to halogen vapors [12]. One way to synthesize electrically conductive polymers is via a chemical polymerization process. Monomers are oxidized by oxidants or catalysts using CuCl_2_, for instance, to form conducting polymers [13]. Among the electrically conducting polymers consisting of five-membered heterocycles, polypyrrole (PPy) has been considered a suitable material. This is due to its high conductivity, simple synthetic process, chemical stability, corrosion resistance, and technological applications [14,15,16]. Moreover, it has already been used in composite materials: with TiO_2_ and with CaCO_3_ [17,18,19]. The synthesis of PPy via oxidative polymerization can be accomplished by first adding the oxidant, copper chloride (CuCl_2_), and then initiating the polymerization by adding Py [18]. Wang et al. have demonstrated that nanostructured conductive polymers exhibit high electrical conductivity [14]. We therefore aim to transfer the nanostructure of seashells to generate nanostructured polypyrrole to potentially produce a material of industrial interest.

High values of tensile strength, as well as good electrical properties, are required, especially in the automotive, furniture, or security industries. This is also important in shoe industries that demand antistatic properties [20]. The envisioned conductive composite material from waste seashells could provide shoe industries with a material that adds to the electrical conductivity with tensile strength. The latter can be provided by seashell particles, the electrical conductivity by polypyrrole and as described above, indeed nanostructured polypyrrole shows particularly high electrical conductivities. Therefore, it is desirable to first develop electrically conductive PPy/CaCO_3_ particles; second, design a simple synthesis route which can be scaled up; and third, use waste seashells.

## 2. Results and Discussion

### 2.1. Synthesis of Electrically Conductive PPy/CaCO_3_ Particles

In this research, seashell powder is used to prepare electrically conductive CaCO_3_ particles. The synthesis pathway is structured as follows: (1) the grinding of the seashells to form seashell particles, (2) chemical and thermal treatment of the seashell particles to remove the organic components, (3) synthesis of polypyrrole using the seashell particles as a template, and (4) doping of the material with iodine. The complete procedure is explained in Section 3, Material and Methods. We used the shells of the clam, *Chamelea gallina*, as the source of biogenic CaCO_3_.

In order to increase the specific conductivity of the PPy/CaCO_3_ material, several synthesis parameters including solvent, size of the seashell particles, seashell particle porosity, gas atmosphere, and temperature were optimized. The resulting PPy/CaCO_3_ materials (PPy/CaCO_3_ M1-M8) with different settings of the parameters are listed in Table 1.

Figure 1 shows a scanning electron microscope (SEM) image of the shell powder obtained after ball milling. In order to investigate the influence of the porosity of the particles, different porous clam particles were used. The porosity of immature and mature seashell particles was determined by Brunauer–Emmett–Teller (BET) measurements. Immature seashell particles (ISP) (D_50_ = 2.63 µm; D_50_: 50% of the sample has this size or smaller sizes.) have slightly higher porosity (0.016 cm^3^;/g) and smaller pore sizes (121.97 Å) than mature seashell particles (MSP) that have a porosity of 0.014 cm^3^/g (D_50_ = 0.44 µm) and a pore size of 161.77 Å. This is due to their higher organic to mineral mass ratio. However, as the shell grows and mineralizes further, the growing aragonite platelets in the nacre, for example, push the silk gel within the chitin layers in front of each other until they meet another platelet, and the protein is then positioned between them. Additionally, there will be more chitin layers in the mature seashell and this makes larger pores (161.77 Å) in relation to the immature seashell, hence the larger averaged pore diameter [21,22]. However, since there are not so many pores, the pore volume is smaller. Since the influence of the particle size on the conductivity of the resulting PPy/CaCO_3_ material is unknown, we prepared different-sized particles by changing the ball milling time. The sizes were determined by laser diffraction. The size distributions are expectedly broad, containing a significant amount of nanosized particles. Therefore, the samples are better described as bimodal with particle sizes of 5.26 µm (area = 42.77% µm) and 531 nm (area = 0.471%µm) for the first batch (ISP 1), and 3.77 µm (area = 29.82% µm) and 105 nm (area = 0.004%µm) for the second batch (ISP 2), as determined by Gaussian deconvolutions of the distribution peaks. The MSPs have a size of 101 nm (area = 17.3% µm).

In order to optimally remove the organic content and make room for the polymer in the clam particles, the sample was first treated with a sodium hypochlorite (NaOCl) solution (5%), then thermally treated (2 h at 300 °C), and finally treated again with a NaOCl solution (5%). Thermogravimetric analysis (TGA) measurements show that there is a mass change of 2.60% in untreated clam particles between 200 °C and 600 °C. This weight loss corresponds to the organic and water content of the clam particles. TGA measurements on treated clam particles show that this organic material was completely removed, allowing the conductive polymer to be deposited/infiltrated. At 600 °C and above, the second thermal process occurs, in which CaCO_3_ decarbonates to CaO. (Appendix A) We also verified that the mineral phase did not change during NaOCl and thermal treatment, using attenuated total reflectance infrared spectroscopy (ATR IR) (Appendix A).

Following the treatment of the clam powder described above, oxidative polymerization of pyrrole (Py) was carried out using clam particles as a template. Due to its high electrical conductivity compared to other electrically conducting polymers, simple synthesis process, and chemical stability, PPy is considered to be one of the most promising conductive polymer materials for the envisioned application in shoe soles. The synthesis of PPy ((C_4_H_2_NH)_n_) via oxidative polymerization can be accomplished by first adding the oxidant, copper chloride (CuCl_2_), and then initiating the polymerization by adding Py (C_4_H_4_NH) [18]:n C_4_H_4_NH + 2n CuCl_2_ → (C_4_H_2_NH)_n_ + 2n CuCl + 2n HCl(1)

That leads to the formation of PPy/CaCO_3_ particles. The PPy/CaCO_3_ particles were doped with iodine in a desiccator [23]:(C_4_H_2_NH)_n_ + 0.2 I → [(C_4_H_2_NH)_n_I_0.2_](2)

We tested doping times from one day to one week. The duration of doping has a significant effect on conductivity. Nevertheless, doping times longer than 7 days are considered unpractical; therefore, we doped for 24 h. (Appendix A) After doping the resulting powder with iodine, the electrical resistance can be measured with a two-point probe resistance instrument. The electrical resistance is the reciprocal of the electrical conductivity. By implementing the distance between the electrodes, the specific electrical conductivity can be calculated.

Since PPy is known to be insoluble in many solvents, the synthesis of PPy with clam particles as templates has been verified by ATR IR and TGA measurements [24]. The shape of the material was investigated with SEM. The prepared material possesses a granular structure of rather similar particle size of 487 ± 70 nm. (Figure 2a) PPy coating on cellulose has a similar structure and size [25]. ATR IR bands of PPy, which was synthesized without a template, can be assigned as follows: (ν_1_) 1521 cm^−1^ stretching vibration of the C-C double bonds and the C-C single bonds; (ν_2_) 1300 cm^−1^ stretching vibration of the C-C single bonds and the C-N bonds; (ν_3_) 1168 cm^−1^ stretching vibration of the C-C single bonds; (ν_4_) 1035 cm^−1^ deformation vibration of the C-H single bond; (ν_5_) 904 cm^−1^ stretching vibration of the C-C double bond and the C-N single bond; (ν_6_) 772 cm^−1^ deformation vibration of the C-H single bond and the N-H single bond, and stretching vibration of the C –H single bond and the N-H single bond [26,27]. There are clear similarities between the peaks of PPy synthesized without a template and PPy/CaCO_3_ composite particles, indicating that PPy was synthesized. ATR IR bands at 1082 cm^−1^, 854 cm^−1^, 712 cm^−1^ and 700 cm^−1^ are characteristic peaks for aragonite, showing the seashell part of the composite material [28]. (Figure 2e,f) The distribution of CaCO_3_ throughout the powder was determined with energy X-ray dispersive spectroscopy mapping, showing that Ca^2+^-ions (red points) are homogeneously distributed throughout the material (Figure 2c,d).

After infiltrating the oxidant, CuCl_2_, into seashell particles, indicated by the green color, two different methods of performing oxidative polymerization were tested. In the first variant, the polymerization took place under air’s atmosphere. In the second variant, the polymerizations were carried out under a nitrogen atmosphere. During the polymerizations of polypyrrole, primary cation radicals combine to form dimers [29]. In order to prevent side reactions, such as the reaction of these radicals with oxygen, the reaction was carried out under a nitrogen atmosphere. The material that was synthesized under nitrogen atmosphere exhibits a specific conductivity of 3.7 × 10^−5^ ± 8.4 × 10^−6^ S/cm (PPy/CaCO_3_ M4). However, PPy/CaCO_3_ M1 which was formed under air’s atmosphere showed a lower specific conductivity (2.8 × 10^−6^ ± 6.0 × 10^−7^ S/cm). In addition, ATR-IR data show that the relative transmittance of PPy/CaCO_3_ M4, which was carried out under nitrogen atmosphere, was lower, which means that the polypyrrole content was higher than that of PPy/CaCO_3_ M1.

Another parameter that was investigated was the number of CuCl_2_ infiltrations. During the synthesis of PPy/CaCO_3_ M4, clam particles were infiltrated once with CuCl_2_, which resulted in a comparable conductivity (3.7 × 10^−5^ ± 8.4 × 10^−3^ S/cm) with those infiltrated twice with CuCl_2_ (PPy/CaCO_3_ M5) (4.0 × 10^−5^ ± 1.31 × 10^−5^ S/cm). Therefore, a single infiltration was considered to be sufficient.

According to the capillary Equation (3), where *h* is the height of a liquid column, *γ* is the liquid–air surface tension, *θ* is the contact angle, *ρ* is the density of the liquid, g is the acceleration due to gravity, and r is the radius of the tube, the wettability of seashell particles can be increased by reducing the contact angle of the solution. A mixture of Py and methanol has a smaller contact angle than a mixture of Py and isopropanol [18]. By improving the wettability of seashell particles with methanol, the ability of Py to enter the nanopores and form polypyrrole not only on the surface, but also in the nanopores is increased [18].
(3)$$h\;=\;\frac{2\gamma \cos(\theta )}{\rho \mathrm{rg}}$$

Indeed, PPy/CaCO_3_ M5 synthesized by adding isopropanol exhibited a lower specific conductivity (4.0 × 10^−5^ ± 1.31 × 10^−5^ S/cm) and polymer amount (34.07%) (Appendix A) than PPy/CaCO_3_ M6 formed by adding methanol (1.2 × 10^−4^ ± 3.3 × 10^−5^ S/cm). Additionally, the polymer content was higher in PPy/CaCO_3_ M6 synthesized in methanol (~52.42%) compared to the material synthesized in isopropanol (Appendix A).

As the size of the seashell particles varied for the same seashell material (pore volume = 0.016 cm^3^/g), different specific conductivities of the resulting composite material were observed. PPy/CaCO_3_ M7 synthesized using IS 2 particles (3.77 µm (area = 29.82% µm) and 105 nm (area = 0.004% µm)) exhibited a specific conductivity of 3.6 × 10^−5^ ± 1.2 × 10^−5^ S/cm. When the size of the clam particles was larger (5.26 µm (area = 42.77% µm) and 531 nm (area = 0.471% µm)), the resulting composite material (PPy/CaCO_3_ M8) had a higher specific conductivity (2.1 × 10^−4^ ± 3.2 × 10^−5^ S/cm). This was counterintuitive. We would have expected that with a smaller size, and thus a larger surface area, there would be more space for polymerization, resulting in higher specific conductivity.

The PPy/CaCO_3_ composite material (PPy/CaCO_3_ M8) with the highest conductivity (2.1 × 10^−4^ ± 3.2 × 10^−5^ S/cm) compared to the other PPy/CaCO_3_ materials (PPy/CaCO_3_ M1-M7) was prepared using seashell particles with sizes of 5.26 µm and 0.531 µm and porosity of 0.016 cm^3^/g, adding the pyrrole mixed with methanol (1:2) and performing the polymerization with pyrrole under nitrogen atmosphere (Figure 3).

To investigate whether the total seashells grain wettability was achieved, contact angle measurements were performed, in which one drop consisting of a mixture of Py and methanol (1:2) was applied on a calcite surface (Appendix A). The surface was completely wetted by the drop. This indicates that the polymerization also took place in the pores of the particles. Additionally, we tested the electrical conductivity at the surface of the particles by measuring the current of the material with nanoscale electrodes at a distance of 21.59 µm. (Figure 4a,b) After measuring the current, we used a linear fit in the low voltage range (+/- 0.2 V) and applied Ohm’s law to determine the resistance. (Figure 4c,d) The specific resistance was 477,139 Ω cm, which corresponds to a specific electrical conductivity of 2.1 × 10^−6^ S/cm. Once the PPy/CaCO_3_ material is pressed to a pellet for measuring the resistance, we had a connection between each particle, resulting in the respective higher electrical conductivity (2.1 × 10^−4^ ± 3.2 × 10^−5^ S/cm). In addition, using BET analysis, we determined a surface area of 10.72 ± 0.15 m^2^/g of the PPy/CaCO_3_ composite material (Appendix A). The surface of the PPy polymerized from solution is comparable with 13.3 m^2^/g [18]. However, the conductivity of another PPy/CaCO_3_ composite material, but with sea urchin spine as a template, produced so far is higher (7.6 S/cm or 3.1 × 10^−2^ S/cm) [18]. Additionally, in comparison with the conductivity of pure polypyrrole, which is in a range between 10 S/cm and 10^−2^ S/cm, the results of this work are outside this range [30]. Nevertheless, they are potentially well suited, as antistatic agents since the range for those agents are from 1 × 10^5^ [Ω] to 1 × 10^9^ [Ω] at 100 V.

### 2.2. Application of PPy/CaCO_3_ Particles as an Antistatic Agent

As a result of their conductivity, PPy/CaCO_3_ particles are potentially ideally suited as an antistatic agent. In this study, PPy/CaCO_3_ M3 particles were added to the sole material Levirex^®^ LX B/282 (Finproject Spa) safety footwear application in a compression molding process. In a preliminary experiment, different percentages (30 wt% and 10 wt%) of the particles were tested as an additive to the sole material. The resulting material (which is shown in Figure 5b,c) is not homogeneous like the standard material. This suggests that the mixing capacity of the two-roll mill opener is not high enough for a good material dispersion and/or that there is not good compatibility between PPy and ethylene vinyl acetate compound (Levirex^®^ LX B/282). Measurements of volumetric resistivity (VR) at 500 V showed that when 30 wt% PPy/CaCO_3_ particles were added to the sole material, the resulting material had a resistivity of 7.0 × 10^7^ [Ω/cm]. When 10 wt% PPy/CaCO_3_ particles were added to the sole material, the resulting material had a resistivity of 1.9 × 10^8^ [Ω/cm]. Both values are within the range for antistatic materials (1 × 10^5^ [Ω] < VR at 100 V < 1 × 10^9^ [Ω]) as the requirement for safety shows according to standard EN 20345-2012 and are comparable to the reference material Levirex^®^ LX B/518 (Finproject Spa) (3.2 × 10^7^ [Ω/cm]) The resulting blend (Figure 4) of the sole material Levirex^®^ LX B/282 with 10 and 30 wt% PPy/CaCO_3_ is not so homogeneous as the standard samples.

Applying PPy/CaCO_3_ powder to Levirex^®^ LX B/518 can cause changes in the mechanical properties and their glass transition temperature. By performing dynamic scanning calorimetry, the glass transition temperature and the melting point were determined (Appendix A). With tensile strength measurements, the tensile strength was measured, elucidating the mechanical property of the material. Comparing the reference material with the material in which PPy/CaCO_3_ powder was added, we could see no change in the glass transition temperature or slight changes in the melting temperature. (Table 2) However, the addition of 10 wt% of PPy/CaCO_3_ to Levirex^®^ LX B/518 decreased the tensile strength (0.701 MPa). In contrast, the tensile strength increased (2.35 MPa) when adding 30 wt% to Levirex^®^ LX B/518). TGA measurements revealed an inorganic filling content of 9.38 % for the material where 30 wt% of PPy/CaCO_3_ was added to Levirex and 5.44 % for the material where 10 wt% of PPy/CaCO_3_ was added to Levirex. (Appendix A)

## 3. Materials and Methods

### 3.1. Preparation of Clam Shell Particles

Clam shells were ground wet in a ball mill (for ISP 1 and MSP: milling: 550 rpm, 30 min. pause: 5 min. cycles: 6. For ISP 2: milling: 550 rpm, 30 min. pause: 5 min. cycles 12) using 2-propanol as solvent. The ball milling was performed with a Planeten-Monomühle Pulverisette 6 classic line by Fritsch. To remove the organic matrix from the shell, the powder was treated with a NaOCl solution (5%) for 24 h and subsequently thermally treated with a muffle furnace L3/12 from Nabertherm with a Controller P320. The samples were each heated for 45 min to 300 °C, held at this temperature of 300 °C for three hours, and then brought to room temperature for 45 min and again treated with a NaOCl solution (5%) for 24 h.

### 3.2. Synthesis of Polypyrrole: Adding Py First

Ground clam particles were weighed in (0.200 g) and pyrrole (0.67 mL, 0.01 mol) was added. The solvents and their amounts were varied. The compositions were: 75% pyrrole and 25% 2-propanol, and 75% pyrrole and 25% methanol. The mixtures were shaken at 4 °C for at least 72 h. After that, the brownish solution was centrifuged, and 4 mL of a CuCl_2_-2-propanol solution (200 mM) was added to the brownish precipitate. The obtained black solution was shaken for 24 h at room temperature. Subsequently, the black precipitate was centrifuged and mixed twice with 6 mL of 2-propanol and once with 6 mL MilliQ water for washing.

### 3.3. Synthesis of Polypyrrole: Adding CuCl_2_ First

The synthesis was carried out under N_2_ gas. First, 0.108 g copper CuCl_2_ was dissolved in 4 mL of 2-propanol or methanol and 0.200 g seashell particles were added. The yellowish solution was stirred for 1 h at room temperature and then evaporated. In the twofold infiltration, 0.108 g of copper chloride and 4 mL of 2-propanol or methanol were added. The mixture was stirred for 1 h at room temperature and then evaporated. 

#### 3.3.1. Polymerization under Nitrogen

The mixture of copper chloride and clam particles was cooled to between −50 °C to −70 °C and pyrrole was added, as well as 2-propanol and methanol, respectively. At these temperatures, the solution was stirred for 30 min and then the mixture, which turned black after completion of the polymerization, was stirred for another 24 h at room temperature. Successively, the solution was centrifuged, and the black precipitate was washed twice with 2-propanol or methanol and once with ultrapure water. The solid was dried on the freeze dryer (ALPHA 2–4 LSC from Christ) for at least 7 h.

#### 3.3.2. Polymerization in the Desiccator

The seashell powder-CuCl_2_ mixture was deposited on silicon wafers and exposed to a pyrrole vapor. For this purpose, the wafers were placed in a desiccator, which also contained a beaker with pyrrole [31].

### 3.4. Doping

For the doping of the solid, the PPy-modified seashell powders were placed in a desiccator (2.15 L) with 0.1523 g of iodine for 24 h and exposed to iodine vapor.

### 3.5. Compression Molding

The blends were prepared by adding the fillers to the standard compound Levirex^®^ LX B/282 (Finproject SpA, Ancarano TE, Italy), which is a crosslinkable and expandible EVA (Ethylene Vinyl Acetate)-based compound. It also contains ADC as a blowing agent and DCP as crosslinking. In order to obtain the blends, a two-roll mill open mixer (G3 s.r.l Mixing Technology, Ancarano TE, Italy) was used at 95 °C for a total mixing time of 10 min for the addition of the additive phase. This temperature was chosen to melt the compound without activating the crosslinking and foaming agents. The obtained still thermoplastic compounds were cut into pieces that were packed in the mold just before closing it, each piece at 90° to the others, to avoid anisotropy of the molded samples due to the orientation of the chains during compounding with the two-roll mill. The cavity of the mold is 140 × 65 × 6 mm. Molding conditions for thermoplastic and compact samples: 95 °C for 480 s, cooling at 30 °C for 10 min and opening of the mold. Molding conditions for crosslinked and foamed compounds: 162 °C for 600 s and opening of the mold.

### 3.6. Characterization

For Brunauer–Emmett–Teller (BET) measurements, the samples were first degassed using a micromeritics Vac Prep 061, and second, the surface area, pore volume, and the pore size were measured on a micromeritics TriStar. The conductivity measurement requires a solid pellet to be pressed from the powdered sample. For this purpose, the MP150-MP250 manual laboratory press from msscientific Chromatographie-Handel GmbH was used. Each sample was pressed for 30 min under a pressure of 5 tons. The specific resistance was measured with two electrodes of a Voltcraft VC220 digital multimeter. The DSC 204 F1 240-10-037-K was used for the differential scanning calorimetry (DSC) measurements. The samples were heated from 35 °C to 180 °C, from 180 °C to −50 °C, and from −50 °C to 180 °C. The heating rate was 10 K/min. The gas flow, which prevails during the measurement, consists of nitrogen. The nitrogen flow is 20 mL/min and 50 mL/min. Energy dispersive X-ray spectroscopy (EDX) was also performed by Gemini 500 operating at 5 kV. Infrared spectroscopy measurements of the samples were performed on an Agilent Cary 630 FITR spectrometer from Agilent Technologies using the ATR setup. Conductivity measurements with nanoprobing tips were performed via five forward (−1 V to +1 V) and backward (+1 V to −1 V) voltage scans to check the stability of 2-point I-V measurements with two tungsten nanoprobing tips (with a diameter of 200 nm) connected to a Keithley 2401, which can be controlled remotely by a Matlab program. Scanning electron microscope (SEM) measurements of the samples were performed on the Carl Zeiss CrossBeam 1540XB microscope. The microscope was used with an accelerating voltage of 5 kV. The size distribution measurements were performed with Mastersizer 3000 by dispersing clam particles in isopropanol and measuring the distribution in isopropanol. The Netzsch STA 449F3 Jupiter was used for the thermogravimetric analysis (TGA) of PPy/CaCO_3_ M1–M8. The samples were heated from 30 °C to 1000 °C. The heating rate was 10 K/min. The gas flow, which prevails during the measurement, consists of oxygen and nitrogen. The oxygen flow is 90 mL/min and the nitrogen flow is 10 mL/min. The NETZSCH TG 209F1 was used for the TGA measurements of the reference material Levirex^®^ LX B/518 and PPy/CaCO_3_ particles (30 wt% and 10 wt%) added to the reference material. The samples were heated from 30 °C to 980 °C. The heating rate was 20 K/min. The gas flow, which prevails during the measurement, consists first of nitrogen from 30 °C to 880 °C, and second of oxygen from 880 °C to 980 °C. For volume resistivity measurements, the molded plaques were cut in half to obtain 3 mm thickness samples and were cut to obtain the specimen according to standard EN 62631-3-1 requirements. The samples were conditioned at 23 °C with 50% relative humidity for 16 h. Then, they were tested to evaluate the volume resistivity following the CEI EN 62631-3-1 standard. Volume resistivity at 500 V was measured using a Keithley model 6517B Electrometer/High-resistance meter with a 8009 resistivity test fixture device. Tensile strength measurements of ISO 527-2 Model specimens were performed with a Zwick Z005/1446 Retroline tC II.

## 4. Conclusions

In summary, we have developed a synthetic strategy for the preparation of a PPy/CaCO_3_ composite material using seashell waste powder as a template. Different parameters such as the size and the porosity of the seashell particles, the infiltration strategy of the oxidant, CuCl_2_, and polymerization strategies were varied. The most successful one was: using seashell particles with sizes of 5.26 µm and 0.531 µm and a porosity of 0.016 cm^3^/g, infiltrating the seashell particles two times with CuCl_2_ solution, mixing pyrrole with methanol, and performing the polymerization at −50 to −70 °C in a nitrogen atmosphere. The prepared composite materials exhibited an electrical conductivity of 2.1 × 10^−4^ ± 3.2 × 10^−5^ S/cm. We demonstrated that for the first time, we have succeeded in forming chemically functionalized biogenic calcium carbonate from seashell powder. However, the conductivity is not in the same range as a sea urchin spine polypyrrole composite (7.6 S/cm or 3.1 × 10^−2^ S/cm) and of pure polypyrrole, which is in the range of 10 S/cm to 10^−2^ S/cm. Nevertheless, potential applications are in the area of antistatic coatings and electrically conductive ink. Volume resistivity measurements of PPy/CaCO_3_ particles molded in a sole material have confirmed that they meet the requirements for antistatic agents in safety footwear.

## Figures and Tables

**Figure 1 ijms-23-07256-f001:**
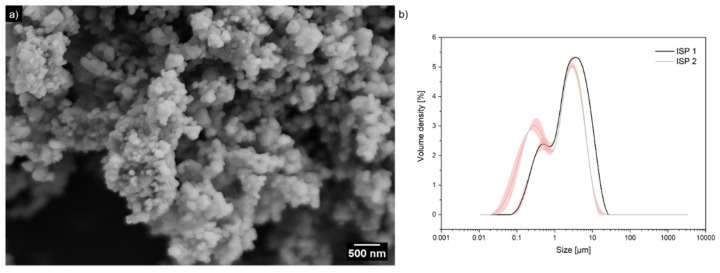
(**a**) SEM image of immature seashell particles (ISP) and (**b**) size distribution of seashell particles: the particle size distribution is obtained using laser diffraction in isopropanol. The graph shows the size distribution of immature seashell particle batches, including the first batch, ISP 1, with particle sizes of 5.26 µm (area = 42.77% µm) and 531 nm (area = 0.471% µm), and 3.77 µm (area = 0.1% µm) and 105 nm (area = 0.004% µm) for the second batch, ISP 2 (gray line). The red area in both curves depicts the error area.

**Figure 2 ijms-23-07256-f002:**
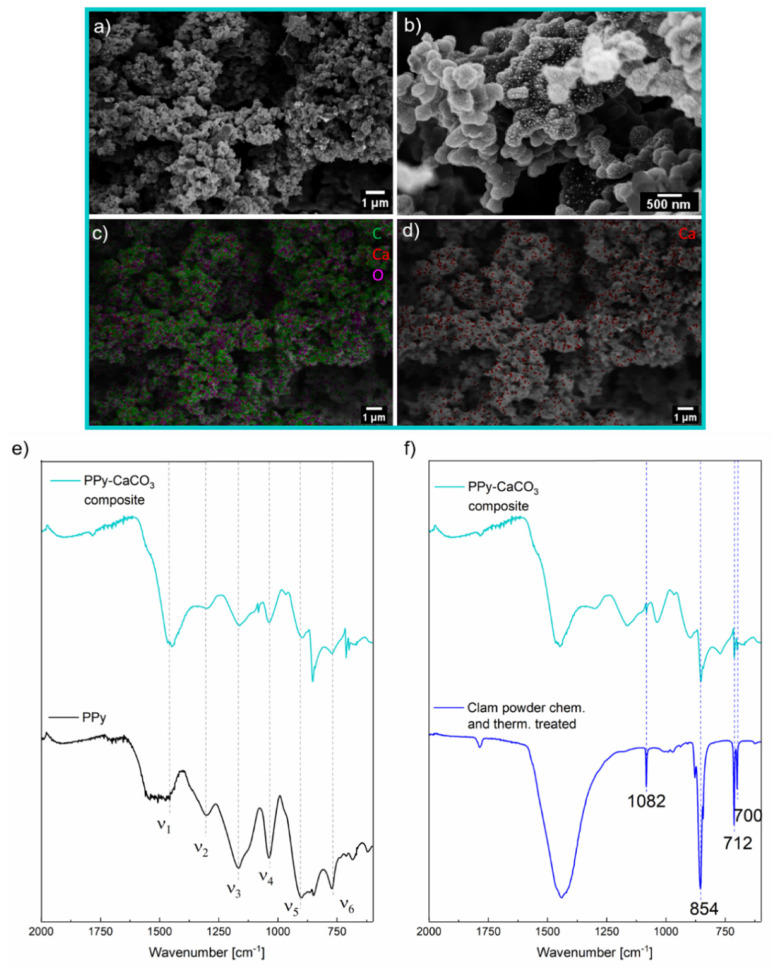
(**a**,**b**) SEM images of a polypyrrole/CaCO_3_ (PPy/CaCO_3_) composite material and in (**c**,**d**) the corresponding EDX mapping images. (**e**) ATR IR spectra of PPy/CaCO_3_ composite (cyan curve) and clam powder (blue curve), which was used for the synthesis of PPy. Bands at 700, 712, 854, and 1082 cm^−1^ are the characteristic aragonite vibrations. (**f**) ATR IR spectra of polypyrrole/CaCO_3_ (PPy/CaCO_3_) composite and PPy synthesized without any template. The dashed lines (ν_1_–ν_6_) indicate the characteristic bands of PPy.

**Figure 3 ijms-23-07256-f003:**
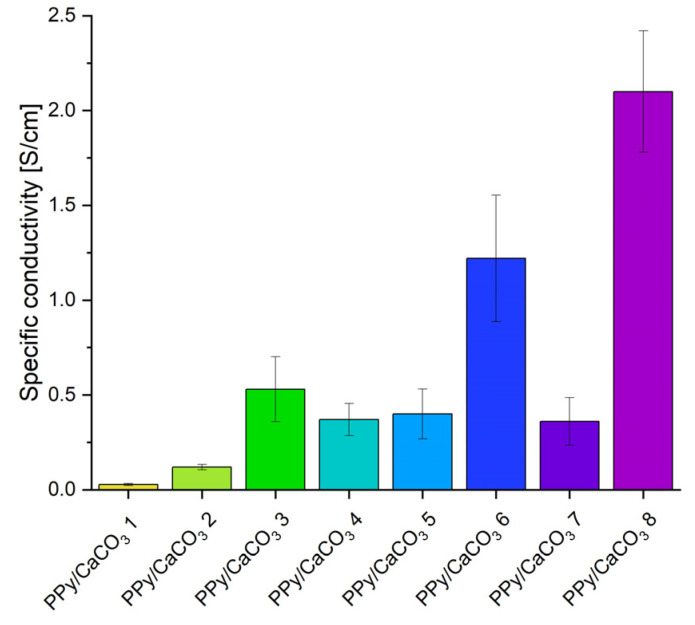
Overview of the specific conductivities of all PPy/CaCO_3_ materials (PPy/CaCO_3_ M1-M8) which are explained in detail in Table 1.

**Figure 4 ijms-23-07256-f004:**
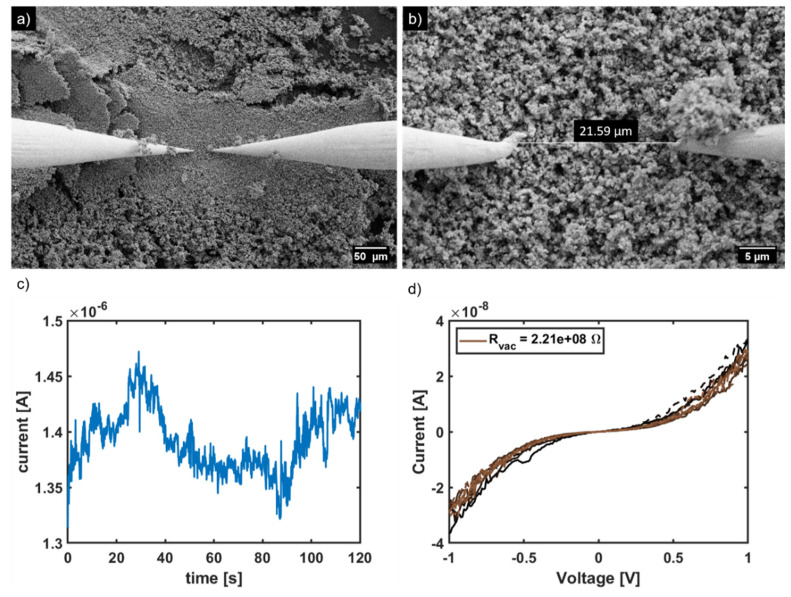
(**a**,**b**) SEM images during measurement of current between two nanoscale electrodes. (**c**) Measured current with time. (**d**) Calculated resistance in the near low voltage regime.

**Figure 5 ijms-23-07256-f005:**
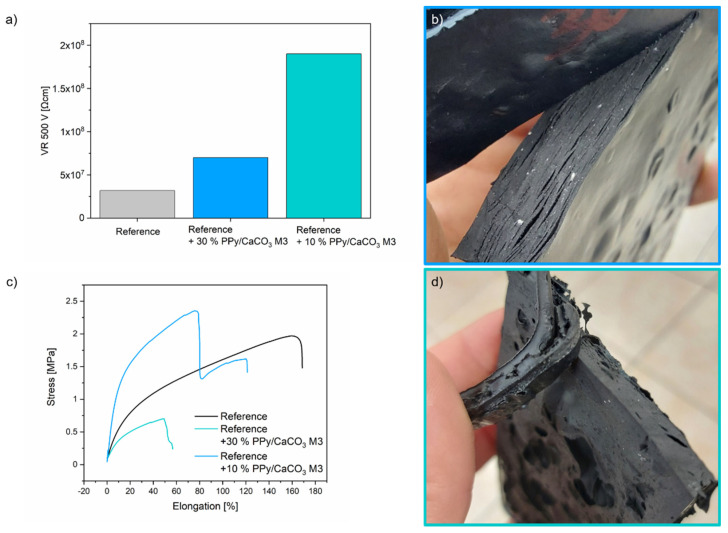
(**a**) Volume resistivity (VR) in Ω cm at 500 V measurements and (**c**) Tensile strength measurements of the reference material Levirex^®^ LX B/518, PPy/CaCO_3_ particles (30 wt% and 10 wt%) added to reference material. PPy/CaCO_3_ particles (30% (**b**) and 10% (**d**)) molded in reference material Levirex^®^ LX B/282.

**Table 1 ijms-23-07256-t001:** Overview of PPy/CaCO_3_ materials (PPy/CaCO_3_ M1-M8) and their respective synthesis parameters including whether the synthesis was performed under a nitrogen atmosphere, solvent, amount of CuCl_2_ infiltrations, the porosity and the size of the clam particles which were used as a template, whether Py or CuCl_2_ added first in the synthesis.

Material	Nitrogen Atmosphere	Solvent	CuCl_2_ Infiltrations	Porosity [cm^3^/g]	Size [µm]	Py or CuCl_2_ First
PPy/CaCO_3_ M1	No	Isopropanol	1×	0.014	0.101	CuCl_2_
PPy/CaCO_3_ M2	No	Isopropanol	1×	0.014	0.101	Py
PPy/CaCO_3_ M3	No	Methanol	1×	0.014	0.101	Py
PPy/CaCO_3_ M4	Yes	Isopropanol	1×	0.014	0.101	CuCl_2_
PPy/CaCO_3_ M5	Yes	Isopropanol	2×	0.014	0.101	CuCl_2_
PPy/CaCO_3_ M6	Yes	Methanol	2×	0.014	0.101	CuCl_2_
PPy/CaCO_3_ M7	Yes	Methanol	2×	0.016	3.77 and 0.105	CuCl_2_
PPy/CaCO_3_ M8	Yes	Methanol	2×	0.016	5.26 and 0.531	CuCl_2_

**Table 2 ijms-23-07256-t002:** Overview of reference sample Levirex^®^ LX B/518, PPy/CaCO_3_ particles (30 wt% and 10 wt%) added to Levirex^®^ LX B/518 regarding their glass transition temperatures (T_G_), melting temperatures (T_M_) and tensile strength.

Material	T_G_ [°C]	T_M_ [°C]	Tensile Strength [MPa]
Reference	−12 to −18	87.2	1.97
Reference +10% PPy/CaCO_3_ M3	−12 to −18	87.6	0.701
Reference + 30% PPy/CaCO_3_ M3	−12 to −18	88.1	2.35

## Data Availability

The data presented in this study are available from the corresponding author upon reasonable request.

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
