# Peer review of "Turning Seashell Waste into Electrically Conductive Particles"

_ijms, 2022, doi:10.3390/ijms23137256_

Round 1

Reviewer 1 Report

The subject of the work is the results of research on the use of marine waste seashells obtained during the processing of seafood as an attractive raw material in the formation of polymer composites showing electrical conductivity, i.e. a non-electrifying material.

The work is interesting but requires some additions before being sent for printing.

  1. In the introduction, it is necessary to supplement the information on the current research using seashells in the creation of composites with a polymer matrix (for example; (Journal of New Materials for Electrochemical Systems 22, 025-031 (2019), International Conference on Artificial Intelligence in Manufacturing & Renewable Energy - 2019, https://doi.org/10.1016/j.matpr.2020.02.818) Enter the appropriate comment.

  1. The applied filler, apart from its influence on the electrical conductivity, causes changes in the mechanical properties of the matrix. For practical applications, the decrease in mechanical strength must be relatively small. Please introduce similar dependencies as shown in Figure 4, but regarding mechanical properties (e.g. changes in tensile and bending strength, as well as stiffness modules). Compare the results with the corresponding reference composite.

  1. Assess and present the interactions at the grain boundary of the filler and the matrix, whether the total seashells grain wettability is actually achieved, whether the polymerization of the monomer also takes place on the filler surface, or whether in this region polymerization is inhibited. Such studies are very important, they should show an improvement in the interfacial interaction when using the filler used as a template during the polymerization reaction.

Author Response

see attached document

Reviewer 2 Report

(13-15) The authors wrote: "Biomaterials such as seashells are intriguing due to their remarkable properties including their hierarchical structure from the nanometer to the micro- or even macroscopic scale. Transferring this nanostructure to polymers can improve their electrical conductivit ”. In the content of the article it is difficult to determine what the building material is in the nano or micro scale of the shell.

(25-66) The entire introduction requires some fine-tuning. There are sentences in it that do not explain anything or you do not know what they mean, e.g.

(49-50) what compounds do the authors have in mind?

(56-57) We therefore, aim to transfer the nanostructure of seashells to polypyrrole - it is still unknown what the authors call 'nanostructure of seashell'.

(63-64) how do the authors want to obtain nano-structured polypyrrole and then how do they intend to measure the conductivity of nano-structured polypyrrole?

(83-85) Why is the discussion lacking an explanation of how pores affect pyrrole deposition? So why was this study done? A very laconic description in the lines (176-179) it is a bit too little on this subject.

(95-96) What and for what purpose did the authors prepare via different times of ball milling?

Fig. 1. Is the image of powdered mature shells the same as the image in this drawing? It is clearly visible that this is not an object that can form a nanometric dispersion (as also illustrated in Figures 4b and 4c)

(118-119) And what happens with the obtained CaO? - lack of  discussion

(163-164) what does the 'relative transmittance of polymerization' mean?

Fig.2a It is a pity that this microscopic image cannot be compared with that shown in Fig. 1a.

In general, there is a lot of research in the article, but it has not been thoroughly analyzed. In the descriptions of the tests performed, there is either a chaos or a complete lack of their discussion.

Author Response

see attached document

Reviewer 3 Report

The paper is devoted for composites with seashell waste processing products preparation and investigations. The topic is generally interesting, however the paper contains unexplained places and need major revisions.

Abstract

” Transferring this nanostructure to polymers can improve their electrical conductivity. ”  It not clear

how insulating particles can improve the electrical conductivity of polymers?

” The polypyrrole/CaCO3 material combines the mechanical strength”  unfortunately, I not find any data about mechanical properties.

The aim of the paper should be more specified.

Page 7 lines 237-239, why milling parameters for particles preparation were selected as indicated in this text? Why for investigations were selected particles with sizes indicated in Fig. 1?

Conclusions, lines 328-329  ”The most successful one was” please explain the sense of these words.

Investigations of composites by other techniques (DSC, broadband spectroscopy, DMA and others) would be very useful.

Author Response

see attached document

Round 2

Reviewer 1 Report

Unfortunately, the response submitted by authors was addressed to Reviewer 1,  - but was in response to questions that were not of my own (possibly this was another reviewer). So it was difficult to judge the changes made to the manuscript related to my comments.Even so, some of these replies were in line with my previous comments, and I also noticed changes to the content of the manuscript that may have been related to my previous disclaimers and directions . However, there are still aspects that have not been fully addressed in the current form of the manuscript. 

Please explain the repeated problem.

Assess and present the interactions at the grain boundary of the filler and the matrix, whether the total seashells grain wettability is actually achieved, whether the polymerization of the monomer also takes place on the filler surface, or whether in this region polymerization is inhibited. Such studies are very important, they should show an improvement in the interfacial interaction when using the filler used as a template during the polymerization reaction.

Reviewer 2 Report

Thank you for the Authors' answers. Overall, the article is good and interesting. It is a report on interesting research carried out.

Author Response

Thank you for your positive assessment

Reviewer 3 Report

Authors make proper corrections according to reviewers remarks and I suggest to publish the paper as it is.

Author Response

Thank you for your positive assessment